Article https://doi.org/10.1038/s41467-022-32919-4

# Fabrication of *Salvinia*-inspired surfaces for hydrodynamic drag reduction by capillary-force-induced clustering

Minsu Kim ®[1], Seunghoon Yoo ®[1], Hoon Eui Jeong ®[2] & Moon Kyu Kwak ®[1] ✉

For decades, bioinspired functional materials have been attracting the interest of many researchers for their remarkable characteristics. In particular, some plant leaves are well known for their inherent super-hydrophobic nature. *Salvinia molesta*, a free-floating aquatic fern, has egg-beater-shaped hierarchical trichomes on its surface of leaves. Due to the unique structure and complex wettability of the hairs, this plant has the ability to maintain a stable thick air layer upon the structure when it is submerged underwater. Often referred to as the "Salvinia Effect," this property is expected to be suitable for use in hydrodynamic drag reduction. However, due to the complex shape of the trichome, currently applied fabrication methods are using a three-dimensional printing system, which is not applicable to mass production because of its severely limited productivity. In this work, artificial *Salvinia* leaf inspired by *S. molesta* was fabricated using a conventional soft lithography method assisted with capillary-force-induced clustering of micropillar array. The fabrication method suggested in this work proposes a promising strategy for the manufacturing of *Salvinia*-inspired hydrodynamic drag reduction surfaces.

In the past decades, many studies in bioinspired engineering have been reported to explore unique functional systems that exist in nature and use them for humans[1–5]. Superhydrophobic surfaces are arguably the most successful and famous examples of bioinspired functional materials[6–10]. Due to the nature of a superhydrophobic surface that is not well wetted by water, the contact area between water and surface can be minimized underwater[11,12]. In this case, a high effective slip length can be induced due to the lubricating layer of the air preserved between the water and the surface[13–17]. Many attempts have been made to utilize a superhydrophobic surface for the hydrodynamic drag reduction (HDR) layer. However, because the superhydrophobic surface which has a low affinity with water loses the air lubricating layer even with small fluctuations, practical application has not been realized yet[18].

The surface of *Salvinia molesta*, a floating aquatic fern, has unique hairs with overall superhydrophobicity and local hydrophilicity as shown in Fig. 1[19]. The hydrophilic terminal cells at the ends of the hairs that are divided into four branches and then clustered together have the effect of attaching to the water surface. Along with it, the gently curved hairs elastically deform against the oscillations of the water surface; thus, the air layer could be stably maintained for a longer period of time[20,21]. The characteristics of the *S. molesta* surface are expected to be applicable to HDR surfaces that minimize the contact area with water while stably maintaining the air layer[22–24].

Considering that the unique three-dimensional (3D) curved structure of *S. molesta* is difficult to fabricate by conventional lithography methods, most of the previous studies have used maskless lithography or a 3D printing system to mimic the structure[25]. In this manner, the structure of *S. molesta* could be successfully mimicked. However, due to the low productivity of the fabrication method, it could not be applied to mass production systems for practical commercialization. Moreover, the local hydrophilicity and elastic behavior

[1]Department of Mechanical Engineering, Kyungpook National University, Daegu 41566, Republic of Korea. [2]Department of Mechanical Engineering, Ulsan National Institute of Science and Technology (UNIST), Ulsan 44919, Republic of Korea. ✉e-mail: mkkwak@knu.ac.kr

of the clustered hairs were not considered in previous research, and no studies have been conducted to evaluate the HDR performance of fabricated *Salvinia*-inspired surfaces.

Meanwhile, capillary-force-induced clustering (CFIC), or capillary-force-induced collapse, is the cause of major defects due to the collapse of the structure with a relatively high aspect ratio in various MEMS fabrication processes by the surface tension of the liquid, such as developer solution[26,27]. The aspect ratio of the structure, the geometric period of the pattern, the elastic modulus of the material, and the surface tension of the liquid are involved in the CFIC behavior[28,29]. By appropriately adjusting these parameters, similar to self-assembly, artificially inducing a uniform CFIC of microstructure to fabricate functional surfaces is possible[30–32].

In the present study, regularly clustered arrays of pillars, such as *S. molesta*, were fabricated by intentionally inducing the CFIC phenomenon. Periodically repetitive pattern structure designed to induce uniform CFIC could be easily replicated through conventional soft lithography using polydimethylsiloxane (PDMS). *Salvinia*-like overall superhydrophobicity and local hydrophilicity at the clustered tips could be achieved through simple solution coating processes. In addition, the applicability of the fabricated *Salvinia*-inspired surface to the HDR functional surface was evaluated via the effective slip length measurement experiment using a rotational rheometer.

## Results and discussion

### Design of the master mold

The dimension of the pillar structure should be carefully designed to achieve sufficient structural stability while still being deformable to the capillary interaction force to induce uniform clustering of the micropillar array[28]. In addition, dimensions, such as the height of the pillar and the distance with neighboring pillars, should be designed for the clustering angle at the tips of pillars to be as high as possible.

Here, PDMS was chosen as the replicating material to prepare the micropillar array template due to its well-established properties, including low modulus[33,34]. Polyvinyl alcohol (PVA) solution was used as a liquid to induce the capillary interaction force. After the PVA solution was dried, a thin PVA membrane could temporarily hold the clustered PDMS pillar tips together[35,36]. The PVA solution was prepared with a low concentration for high surface tension, which could induce efficient tip clustering[37].

To induce the intended uniform clustering of micropillars, disruption of the micropillar arrangement due to structural collapse should be avoided. Firstly, the micropillar should be stable against self-collapse due to the gravity and adhesion between the ground and adjacent pillars. For stability against gravity-induced buckling, the critical aspect ratio of the structure against buckling $(\frac{h}{d})_{cb}$ is given as

$$\left(\frac{h}{d}\right)_{cb} = \left(\frac{0.49E}{\rho g d}\right)^{1/3} \quad (1)$$

where $h$ and $d$ are the height and diameter of the pillar, respectively; $E$ is the elastic modulus, $\rho$ is the density of the material; and $g$ is the gravitational acceleration[38]. For stability against ground collapse and lateral collapse, the critical aspect ratio against ground collapse $(\frac{h}{d})_{cg}$ and lateral collapse $(\frac{h}{d})_{cl}$ are respectively given as

$$\left(\frac{h}{d}\right)_{cg} = \frac{\pi^{5/3}}{2^{11/3}3^{1/2}}(1-\upsilon^2)^{-1/6}\left(\frac{E}{2\gamma_{SV}}\right)^{2/3}d^{2/3} \quad (2)$$

$$\left(\frac{h}{d}\right)_{cl} = \left(\frac{3^{3/4}\pi E w^{3/2}}{2^{1/4}32\gamma_{SV}(1-\upsilon^2)^{1/4}d^{1/2}}\right)^{1/3} \quad (3)$$

where $w$ is the separation between adjacent pillars; and $\upsilon$ and $\gamma_{SV}$ are the Poisson's ratio and surface energy of the material, respectively[39,40].

The critical conditions at which a collapse type makes the pillars more vulnerable than the others could be obtained as follows:

$$\left(\frac{h}{d}\right)_{cb}/\left(\frac{h}{d}\right)_{cl} = \left(\frac{0.49 \cdot 32 \cdot 2^{1/4}}{3^{3/4}\pi} \cdot \frac{\gamma_{SV}(1-\upsilon^2)^{1/4}}{\rho g} \cdot \frac{1}{w^{3/2}d^{1/2}}\right)^{1/3} = 1 \quad (4)$$

$$\left(\frac{h}{d}\right)_{cl}/\left(\frac{h}{d}\right)_{cg} = \left(\frac{128 \cdot 2^{3/4}3^{9/4}}{\pi^4} \cdot \frac{\gamma_{SV}(1-\upsilon^2)^{1/4}}{E} \cdot \frac{w^{3/2}}{d^{5/2}}\right)^{1/3} = 1 \quad (5)$$

For PDMS, by substituting the material properties into the above equations, the critical conditions for collapse vulnerability could be simplified as follows:

$$d \approx 4.93 \cdot 10^{12} \cdot \left(\frac{1}{r_c}\right)^3 \quad (6)$$

$$d \approx r_c^{3/5} \quad (7)$$

For three-subpillar design as shown in Fig. 2a, b, the separation between adjacent pillars $w$ is equal to $\sqrt{3}r_c$, where $r_c$ is the center-to-center radius[41].

As shown in Eqs. (6) and (7), the material properties, the diameter, and the lateral separation of the pillars determine the type of collapse that the pillars are most vulnerable to. Equation (6) shows the critical condition in which micropillars become more susceptible to either type of collapse, buckling or lateral collapse. Equation (7) represents the critical condition in which micropillars become more susceptible to either lateral collapse or ground collapse. The diagram depicting these equations and regions of the most vulnerable collapse type is shown in Fig. 2e, and Supplementary Fig. 1. Making the design point lies in the lateral collapse region so that only the stability against lateral collapse should be considered is desirable to make micropillars sufficiently stable against any type of collapse but vulnerable to capillary interaction force.

Structural deflection of the micropillar against capillary interaction force should be considered to ensure regular clustering of the pillar arrays[28,42]. The capillary interaction force at the tip of the pillar should be higher than the restoring force of the pillar in the range up to the maximum deflection, that is, the distance from the original point to the intended clustering point, $r_c$. The capillary interaction force between partially immersed identical pillars is given as

$$F_c \approx \frac{\pi\gamma_{LV}d_t^2\cos^2\theta_r}{2w} \quad (8)$$

where $\gamma_{LV}$ is the interfacial surface energy of the liquid, $d_t$ is the diameter of the micropillar tip, and $\theta_r$ is the receding contact angle[27]. In the case of the three-pillar design shown in Fig. 2a, b, the capillary interaction force $F_{cc}$ exerting on a tip of a subpillar to the center of the lower pillar

$$F_{cc} \approx \frac{\pi\gamma_{LV}d_t^2\cos^2\theta_r}{2} \cdot \frac{1}{r_c - \delta} \quad (9)$$

should be at least higher than the restoring force of the pillar,

$$F_E = \left(\frac{3\pi E d_t d_b^3}{64h^3}\right) \cdot \delta \quad (10)$$

in the region of $0 < \delta < r_c$[28,43,44], where $d_b$ is the diameter of the micropillar base. Typical cases of capillary force and elastic restoring force according to the deflection are shown in Supplementary Fig. 2.

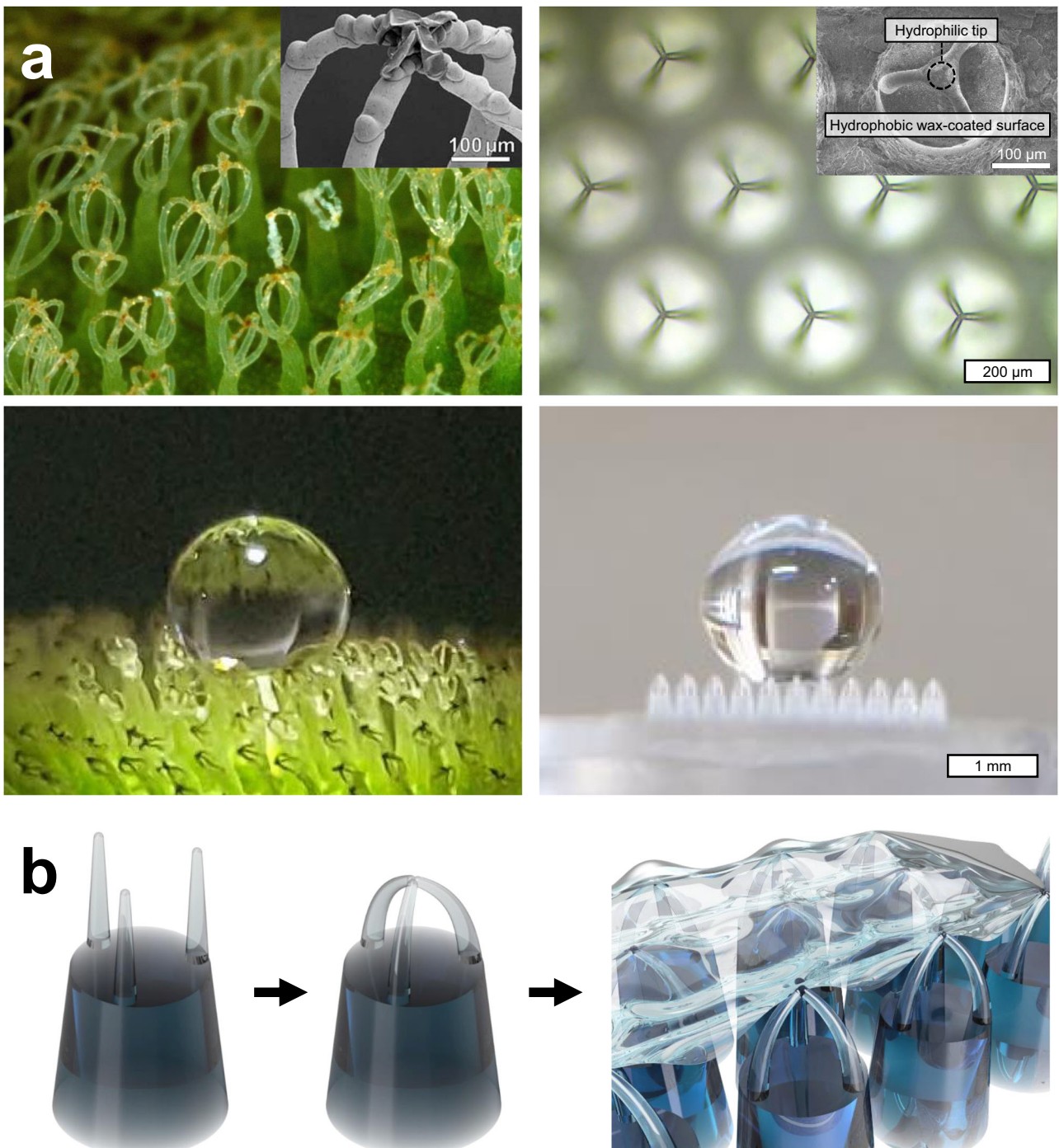

**Fig. 1 | Overall concept of the research. a** Hairs on the *Salvinia* leaves (left) and *Salvinia*-inspired hierarchical microstructures (right). Reproduced with permission[28]. B. Rice, sarracenia.com, Bugwood.org. Copyright 2010, Wiley. **b** Schematics showing the concept of the fabrication of *Salvinia*-inspired HDR surfaces.

The critical condition for $F_{cc}$ to be higher than $F_E$ could be acquired from a discriminant

$$\frac{h^3}{r_c^2} > \frac{3Ed_b^3}{128\gamma_{LV}d_t\cos^2\theta_r} \qquad (11)$$

The critical condition plot is shown in Fig. 2f, and Supplementary Fig. 3.

In addition, to maximize the water repellency of the surface, the clustered pillar array should have gentle curvature, ideally making a horizontal angle to the water at the clustered tips. With the additional contact area per height difference due to the gently bent upper pillars, the energy required for the water to penetrate into the clustered region increases[19]. The angle at the clustered pillar tips could be described implicitly as

$$\frac{r_c}{h} = \int_0^{\varphi_0}\frac{\sin\varphi}{\sqrt{\sin\varphi_0-\sin\varphi}}d\varphi \Big/ \int_0^{\varphi_0}\frac{1}{\sqrt{\sin\varphi_0-\sin\varphi}}d\varphi \qquad (12)$$

where $\varphi_0$ is the angle at the deflected tip[45]. The higher clustering angle $\varphi_0$, the lower the angle between the clustered pillars and the

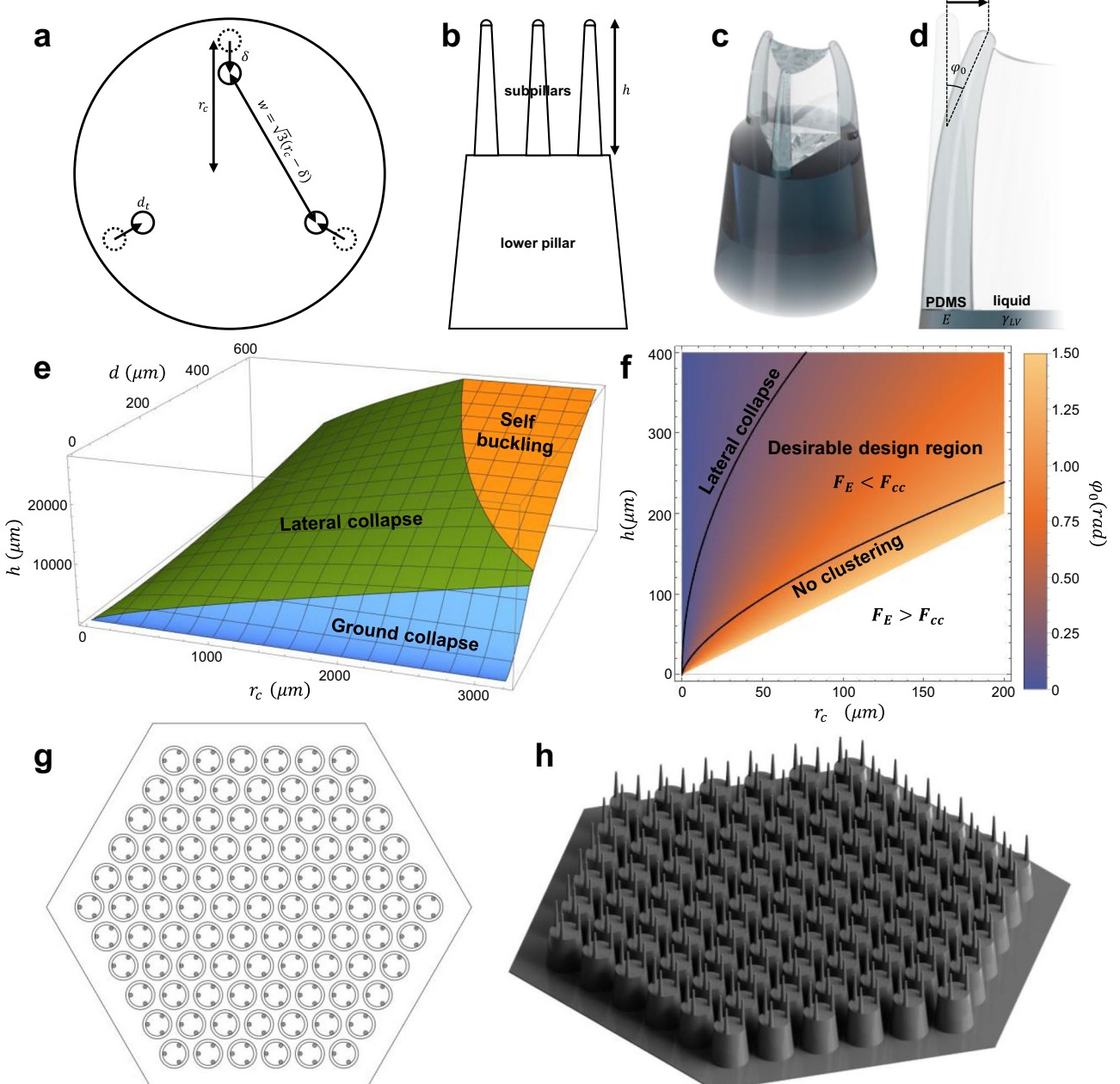

**Fig. 2 | Design of the master mold. a, b** Schematics of major design variables for CFIC, where $d_t$ is the micropillar tip diameter, $h$ is the height of the subpillar, $r_c$ is the center-to-center radius, and $\delta$ is the deflection of a pillar. **c, d** Schematics of a hierarchical pillar being bent to capillary interaction force when partially immersed in a liquid. **e** Three-dimensional plot of the critical conditions for each collapse type. **f** Plot depicting critical conditions for CFIC and clustering angle of the pillar tips. The design point should lie in the desirable design region to induce the clustering of upper pillars while avoiding lateral collapse. **g** Top wireframe view of the designed hierarchical pillar array template. **h** Isotropic view of the designed pillar array template.

water surface. The clustering angle as a function of the pillar height and center-to-center radius is depicted in Fig. 2f.

To sum up, to effectively mimic the "Salvinia Effect" by utilizing CFIC, the pillar should be sufficiently stable to prevent self-collapse but sufficiently flexible to be clustered together to the capillary interaction force, making a minimum angle with the water-air interface by increased clustering angle. An appropriate design point satisfying all these conditions should be slightly above the clustering critical condition plot shown in Fig. 2f.

With desirable design parameters chosen by considering the above conditions, hexagonal periodic arrays of hierarchical micropillar were designed, as shown in Fig. 2g, h. The pillars were arrayed with periods of 350 μm for lower pillars and center-to-center radius

of 100 μm for upper pillars. The lower pillars have a low aspect ratio to grant structural stability while granting more height to upper pillars. The upper pillars have a relatively high aspect ratio to be easily bent, clustered, and then form a similar structure to the hairs of *S. molesta*. The lower and upper pillars were tapered with angles of 5.8 and 3 degrees, respectively, for easier release in the replication step and higher curvature at the tips of the upper pillars when clustered. The upper pillars have height of 220 μm, base diameter of 40 μm, and tip diameter of 17 μm. The distance among three upper pillars on a lower pillar was slightly closer than that with the others for the capillary force to be directed towards the center of the lower pillar. The hierarchical pillar structures were arrayed hexagonally, as shown in Fig. 2g, h, to induce symmetric, regular capillary interaction

force on each pillar. Unlike living *S. molesta*, which has four upper hairs divided from a stem, three upper pillars were assigned for each lower pillar. Regularly arranging the pattern was easier with three upper pillars rather than four. The hierarchical pillar pattern was designed without any overhang or complex part to make it easy to replicate the structure with the conventional lithographic replication process.

### Fabrication of the *Salvinia*-inspired surfaces by capillary-force-induced clustering

From the design, a positive master mold was fabricated using 2PP 3D printing system. The master mold went through surface passivation process for easier release. Samples for the test were prepared through two successive soft lithography steps. First, the PDMS precursor was cast on the master mold, desiccated, and cured in 70 °C oven for 2 h before being demolded to prepare a negative type mold. The negative type PDMS mold underwent additional overcuring in 150 °C oven for 2 h for defect-free self-replication. From the negative type PDMS stamp mold, the positive PDMS replica was prepared by following the same typical soft lithographic process. PVA solution was cast on the as-prepared positive PDMS template and then evaporated in 70 °C convection oven for 2 h. During the evaporation of the PVA solution, the tips of the three upper pillars were clustered together at the center due to the capillary interaction force of the solution, as shown in Fig. 3. The three upper pillars clustered together were temporarily fixed together due to the thin PVA film formed on their hemispherical tips. Given that the PVA film holding the clustered pillars together is fragile, the PDMS precursor was additionally inked and cured at the clustered tips to permanently fix the clustered tips together. Then, the sample was coated with a hydrophobic wax solution to make the entire surface superhydrophobic. The clustered tips were additionally coated with polydopamine (PDA) by making the surface partially submerged by the PDA solution to grant local hydrophilicity. In this procedure, only the clustered tips made direct contact with the PDA solution due to the superhydrophobicity of the surface. Schematics and microscopic images of the entire fabrication process are shown in Fig. 4, and Supplementary Figs. 4, 5.

### Characterization of the fabricated surfaces

Scanning electron microscopic images of the fabricated sample are shown in Fig. 5. The clustering angle of micropillars after the CFIC was measured to be $54.3 \pm 1.5°$, which is in a good agreement with the theoretical expectation from Eq. (12). The hydrophilic tip part is visually noticeable, while the other parts are covered with hydrophobic nano wax crystals. As shown in Supplementary Fig. 6, the fabricated sample surface shows superhydrophobicity with high water contact angle of $174.9° \pm 3.2°$.

Due to the PDA coated hydrophilic tips, the clustered hairs of the fabricated sample pulled the meniscus of water similarly to *S. molesta*, as shown in Fig. 6. By assuming the shape of the meniscus is elliptical as shown in Supplementary Fig. 7, the minimum interfacial energy $W$ required for the meniscus to form is

$$W = A\sigma + \pi\left(\frac{c}{2}\right)\sigma^* - \pi\left(a + \frac{c}{2}\right)^2\sigma \quad (13)$$

where $A$ is the interfacial area of the meniscus; $a$ and $c$ are the elliptical fitting parameters; and $\sigma$ and $\sigma^*$ are the surface tension of the liquid and the interface tension of the contact area, respectively[20]. As the meniscus pulling force is equal to its water adhesion force, it could be calculated as

$$F = \frac{\partial W}{\partial b} = \frac{\partial A}{\partial b}\sigma \quad (14)$$

The calculated meniscus pulling force of a single clustered hair was 19.9 μN, comparably similar to that of *S. molesta* (19.5 μN)[20]. As provided under Supplementary Fig. 7, the meniscus pulling force can also be calculated using different method, which gives values in a similar range[46–48].

Given the meniscus pulling force and number of clustered hairs per unit area, the maximum outward pressure oscillation of the water-air interface the fabricated sample could hold without breaking a pin could be assumed as ~217 Pa.

$$\frac{\eta_{flat}}{\eta_{slip}} = \frac{\tau_{flat}}{\tau_{slip}} = \frac{M_{flat}}{M_{slip}} = 1 + \frac{b_{slip}}{h} \quad (15)$$

In general, an effective slip length of the surface is used to quantitatively compare the effect of HDR[18]. The effective slip length of the prepared *Salvinia*-inspired surfaces was measured using a rotational rheometer. Apparent viscosity of 40 wt% glycerol was measured on the sample surface and the reference plate to calculate the effective slip length to quantitatively evaluate the HDR effect of the fabricated sample, as shown in Fig. 7. From Eq. (15), the effective slip length was found to be ~137 μm[49]. For a torsional Couette flow, by using the measured apparent viscosities from the rheometer test, the drag reduction on the sample surface could also be calculated as ~82.0% from the following equation:

$$DR = 1 - \frac{\eta_{slip}}{\eta_{flat}} = \frac{b_{slip}}{h + b_{slip}} \quad (16)$$

A hierarchical micro-pillar array was designed, replicated by soft lithography, and uniformly clustered by inducing capillary force. Afterward, through hydrophobic and hydrophilic coating processes, the heterogeneous wettability of *Salvinia* was also successfully mimicked. Due to its gently curved structure and local hydrophilicity, the *Salvinia*-inspired surface exhibited high meniscus pulling force to maintain a stable air-water interface while minimizing the water–solid contact area. By using a rotational rheometer test, a high level of effective slip length was measured on the *Salvinia*-inspired surface. Given that the hierarchical micropillar template for the CFIC could be prepared by conventional soft lithography processes, which is easy to apply to a continuous fabrication process, it is expected to contribute to the commercialization of HDR surface.

## Methods
### Master mold fabrication

The original master mold with hierarchical pillar arrays was 3D printed by Photonic Professional GT (Nanoscribe GmbH) by using IP-S resin on an ITO coated glass, with 1 μm resolution. After the fabrication, the mold went through a passivation process to ensure defect-less release in the following soft lithography processes.

### Soft lithography

Templates for the capillary-force-induced clustering were prepared following a typical soft lithography protocol by using PDMS (Sylgard 184, Dow Corning) with 10:1 base to curing agent ratio. PDMS precursor was poured onto the master mold and cured in 70 °C oven for 2 h before demolding to prepare negative type replica mold. The negative type, first-generation PDMS replica mold was baked in 150 °C oven for 2 h for defect-less release in the following self-replication. By using the first-generation replica as a mold, the positive type, second-generation PDMS replica was replicated following the same typical protocol and then used as a template for CFIC.

**Fig. 3 | Microscopic images of micropillars clustering as PVA solution dry out.** **a** Micropillars at their initial position with pillar tips partially immersed by PVA solution. **b** Micropillars being bent due to the capillary force of PVA solution. **c** Micropillar tips making contact with one another. **d** Clustered micropillars right after the PVA solution dried off the surface.

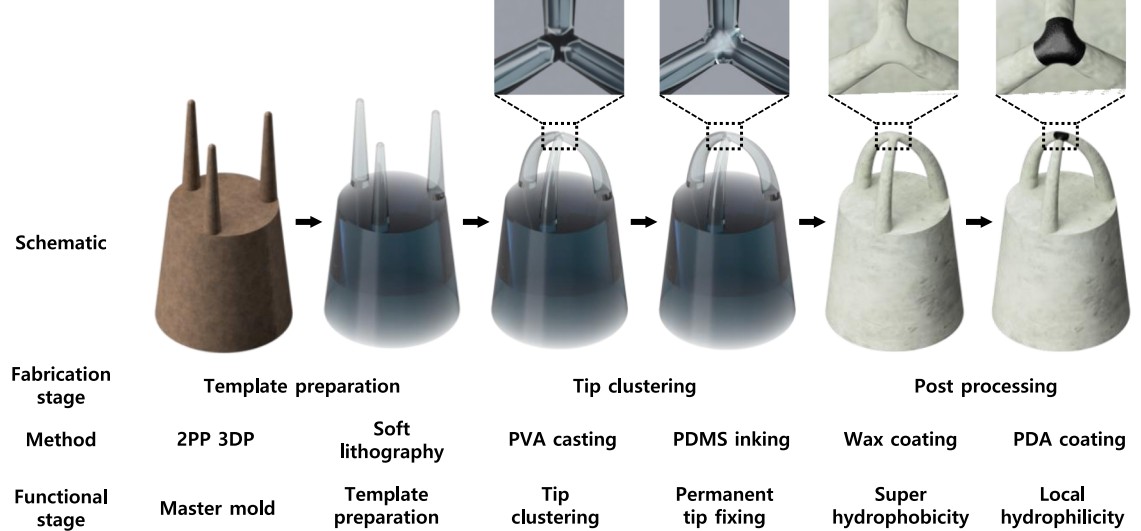

**Fig. 4 | Schematics of the hierarchical pillar structure at each fabrication step.** Each schematic shows the hierarchical pillar structure at each fabrication step, from the original master mold fabricated by 2PP 3D printing system (leftmost) to the final step (rightmost). Insets show schematics of the tip part of the clustered micropillars.

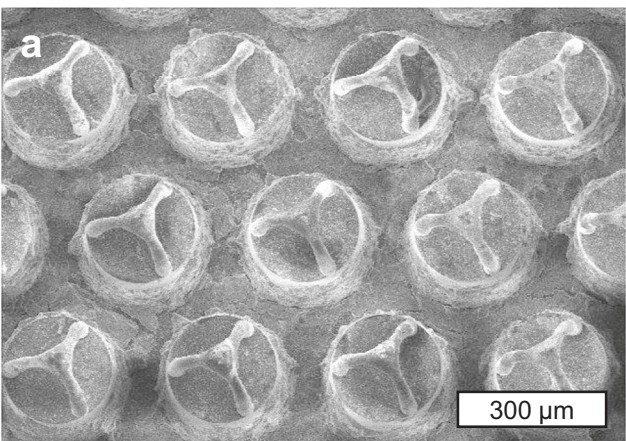
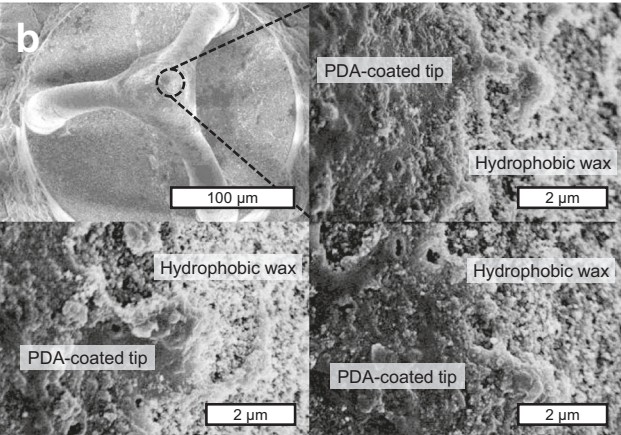

**Fig. 5 | SEM images of the fabricated sample surface. a** Clustered hierarchical pillar arrays. **b** Hydrophobic wax and hydrophilic tips on a single clustered structure.

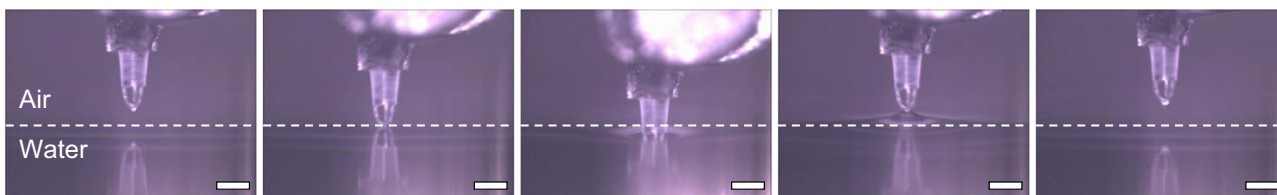

**Fig. 6 | Microscopic images of a single clustered structure pushing and pulling the water meniscus.** *Salvinia*-like behavior with the water-air interface is shown when a single clustered structure of the fabricated *Salvinia*-inspired surfaces comes into contact with water. Scale bars represent 300 µm.

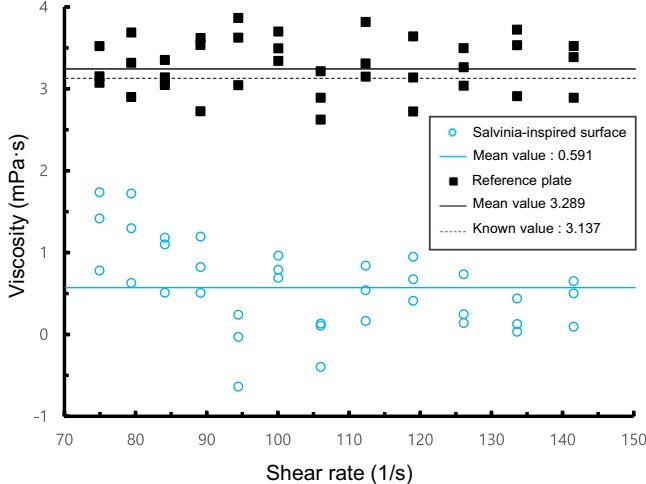

**Fig. 7 | Apparent viscosity of 40 wt% glycerol on the reference plate and the *Salvinia*-inspired surfaces.** Apparent viscosity of 40 wt% glycerol measured by the rotational rheometer on the reference plate (black squares), *Salvinia*-inspired surfaces (blue circles). Mean values for each case are indicated by solid lines, and the known value of the liquid is indicated by the dashed line.

## Capillary-force-induced clustering

PVA solution (0.1 wt%) was prepared by mixing DI water with PVA (Sigma Aldrich) powder. The PVA solution was applied on the patterned area of the as-prepared second-generation PDMS replica. After bubbles were removed, the PDMS template with PVA solution on it was placed in a 70 °C convection oven until the solution dried out completely.

## Fixing the clustered tips with additional PDMS

The clustered tips were carefully inked by additional PDMS. The inked PDMS was fully cured in a 70 °C oven to fix the clustered tips in the position.

## Additional hydrophobic and hydrophilic coatings

The entire sample surfaces were coated with commercial hydrophobic wax solution (Ultra ever dry–top coat, UltraTech International, Inc). The was solution was applied to the entire surfaces. After drying, excessive wax was blown off the surfaces by compressed air. To make the clustered pillar tips hydrophilic, the clustered tips were coated by PDA by leaving the tips in contact with the DA solution for 24 h. The DA solution was prepared using dopamine hydrochloride (Sigma Aldrich) and Tris-HCL buffer solution at pH 8.5.

## Rheometer test

A rotational rheometer (HAAKE MARS, Thermo Fisher Scientific) was used to measure the apparent viscosity of 40 wt% glycerol using a plate probe with a diameter of 20 mm. The apparent viscosity of the solution was measured on the reference plate and *Salvinia*-inspired sample surfaces. Multiple *Salvinia*-inspired surfaces samples were put together to fully cover the probe area. The distance between the measuring surfaces and the probe was controlled to be 30 µm. 10 ml of the solution was applied carefully between the sample surface and the rotating probe by using a syringe.

## Data availability

The data that support the findings of this study are available within this paper and or included in the Supplementary Information, and from the corresponding authors upon request. Source data are provided with this paper.

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

## Acknowledgements

This work was supported by the National Research Foundation of Korea grant (NRF-2020R1A4A1018652, M.K.K. and 2019R1A2C1086766, S.Y.) and Bridge program by the Korea Environmental Industry & Technology Institute (2021002800015, M.K.), which was funded by the Korean government. The authors would like to thank Dr. Misato Iwashita at Korea Brain Research Institute for the kind help with the rotational rheometer test.

## Author contributions

M.K. conceived the project. M.K.K. and H.E.J. supervised the project. M.K. designed the master mold. M.K. performed 2PP 3D printing of the master mold. M.K. and S.Y. carried out soft-lithographic replications and surface modifications. M.K. performed capillary-force-induced clustering of micro pillar arrays. M.K. performed additional hydrophilic and hydrophobic coatings on the fabricated sample. M.K. performed all tests for characterization. All authors discussed the results and commented on the manuscript.

## Competing interests

The authors declare no competing interests.
