## [Peer Review File · Nature Communications]

REVIEWER COMMENTS

Reviewer #1 (Remarks to the Author):

In the present manuscript, the authors propose the fabrication of a biomimetic surface inspired by *Salvinia molesta* leaves. They exploited a conventional soft lithography-based fabrication method followed by capillary-force-induced clustering of micropillars. They were able to reproduce the chemical functionalization present on the natural hairs. In this way, the authors were able to obtain hierarchical microstructures responsible for super-hydrophobicity and drag reduction. I think that the noteworthy result of the study relies on the possibility to obtain 3D crown-like shapes with a relatively simple procedure. For this reason, the manuscript is of significance to the field of biomimetic wettability and drag reduction.

The authors provided also a theoretical description of the involved physical phenomena, supporting the chosen design. Finally, the hydrodynamic performance of the proposed patterned surface has been evaluated via effective slip length.

The paper presents several critical points that in my opinion should be properly addressed before publication:

- 1) The geometrical characteristics of the microstructures are not completely reported, some of them are missing (center-to-center radius, micropillar height and radius...). Also, other physical values should be reported like density, elastic modulus, etc.
- 2) Rotational rheometer test should be better described, in particular with respect to the dimension of the tested sample. In fact, what is the dimension of the sample that the authors tested?
- 3) Regarding the dimension of the patterned surface, what is the rate of success of the CFIC and the chemical functionalization?
- 4) Line 146: the authors state that the appropriate design point satisfying all the conditions for effective CFIC should be slightly above the clustering region critical condition where the angle with the water-air interface is minimum (line 144). Anyway, this seems in contrast with figure 2f where the angle is minimum far from the clustering region critical condition.
- 5) Line 189: the authors should explain in a better way how they performed the chemical functionalization. Moreover, how long does it last?
- 6) Authors state that their strategy is promising for the practical commercialization of the *Salvinia*-inspired HDR surfaces: is the high aspect ratio of the micropillars achievable also with other microfabrication techniques or is it somehow "limited" by the employment of two-photon lithography?

Some more aspects need to be clarified: here below are a few recommendations that authors can consider to revise:

- 1) Line 135: the angle at the clustered tips should be illustrated in a picture in order to clarify this concept.

- 2) Figure 4: the phases of the fabrication process should be illustrated rather than, or not only, the images of the results.
- 3) Line 250: since the authors state about the gentle curvature of the structures, did they measure it?
- 4) Equation 4,5,6,7: the subscript letters of the geometrical parameters should be better explained before the equations.
- 5) Equation 15: it should be reported later in the text.

Reviewer #2 (Remarks to the Author):

This is a beautiful piece of work, well conceived and well worthy of publication.

My few criticisms/comments are as follows.

Line 92 et seq.

The description of Equations (1) – (5) is unclear. Although references are given, there is insufficient explanation in the context.

Equation (1) describes self buckling due to gravity. This seems to be basically Euler's formula for the critical load leading to buckling of the beam under its own weight. It would seem to suggest that buckling will not (usually) occur due to gravity, but will, with an applied force and sufficiently large (h/d) . However, consideration of polymer surface tension may be equally relevant at this micro-scale. Any bending of the pillar, assumed for simplicity to be a circular cylinder, will induce an effective resistance due to surface tension, γ , of order $\gamma\pi d$. The two effects, elastic and capillary, should be approximately additive and so using the Euler expression, where $I = \pi d^4/64$, the buckling force should be of order $P = \pi^2 EI/h^2 + \pi\gamma d$, where the two components are of similar magnitude for the given system.

Following this, Equation (2) and (3) describe 'ground collapse' and 'lateral collapse'. This is unclear, at least to me. (Although I do note the presence a surface tension term in both.)

Line 174 et seq.

There is a description of the fabrication process mentioning the 'PDMS precursor', '...negative PDMS stamp, the second positive PDMS replica was demolded with the same typical process. PVA solution was cast on the as-prepared positive PDMS sample and then evaporated...', etc. For me, at least, it is unclear what polymer the final structure is made of!

The authors used polyvinyl alcohol (PVA) solution as a liquid to induce the capillary interaction force.'...Why specifically PVA?

Line 209 et seq.

An elliptic approximation is adopted for the meniscus shape in Equation (13). Presumably this is for simplicity, but the solution for such a problem has been known for years. The effective 'weight' of the meniscus, which amounts to the pull-off force, is simply $C\sigma \cos \theta$, where C is the circumference of contact, σ liquid surface tension, and θ the local contact angle. See for example: Dynes and Kaelble (J. Adhesion, 1974), and Schultz et al. (J. Adhesion, 1981). Admittedly, in the present case, it is difficult to determine either C or θ ! However, provided the solid is homogeneous, axially circular and its axis normal to the free liquid surface (Alimov and Kornev (Proc. Roy.Soc. A, 2014)), the FORM of the meniscus is independent of the solid. See, for example, White and Tallmadge (J. Fluid Mech., 1965).

A couple of minor points:

Supplementary Information, Figure 5, cites contact angle as 168.3° . Are 4 significant figures warranted, especially with no estimate of errors? Also, do the authors know what this contact angle corresponds to, since the drop is 'sitting' on an inhomogeneous bed of pillars, or to coin Quéré (Nature Materials, 2002), it is a 'Fakir droplet'.

There seems to be a typo. in Figure 2. There appears to be no d (as in the legend), but there is a dt .

Point-to-point responses to the reviewers' comments

Reviewer #1:

In the present manuscript, the authors propose the fabrication of a biomimetic surface inspired by *Salvinia molesta* leaves. They exploited a conventional soft lithography-based fabrication method followed by capillary-force-induced clustering of micropillars. They were able to reproduce the chemical functionalization present on the natural hairs. In this way, the authors were able to obtain hierarchical microstructures responsible for super-hydrophobicity and drag reduction. I think that the noteworthy result of the study relies on the possibility to obtain 3D crown-like shapes with a relatively simple procedure. For this reason, the manuscript is of significance to the field of biomimetic wettability and drag reduction.

The authors provided also a theoretical description of the involved physical phenomena, supporting the chosen design. Finally, the hydrodynamic performance of the proposed patterned surface has been evaluated via effective slip length.

The paper presents several critical points that in my opinion should be properly addressed before publication:

Q 1: The geometrical characteristics of the microstructures are not completely reported, some of them are missing (center-to-center radius, micropillar height and radius...). Also, other physical values should be reported like density, elastic modulus, etc.

Answer. Dimensions of micropillars are clarified in lines 167 ~ 168. Other physical values are added in Supplementary table 1.

Q 2: Rotational rheometer test should be better described, in particular with respect to the dimension of the tested sample. In fact, what is the dimension of the sample that the authors tested?

*Answer. A rotational rheometer (HAAKE MARS, Thermo Fisher Scientific) was used to measure the apparent viscosity of 40 wt.% glycerol using a plate probe with a diameter of 20 mm. The apparent viscosity of the solution was measured on the reference plate and salvinia-inspired sample surfaces. Since the hexagonal pattern area of the salvinia-inspired sample is $\sim 9.88 \text{ mm}^2$, which is smaller than the probe, multiple samples were meticulously put together to fully cover the probe area while minimizing the seam between samples. The distance between the measuring surfaces and the probe was controlled to be $30 \mu\text{m}$. 10 ml of the solution was applied carefully between the sample surface and the rotating probe by using a syringe. Relevant context is added in **Methods** section (Lines 301 ~ 307).*

Q 3: Regarding the dimension of the patterned surface, what is the rate of success of the CFIC and the chemical functionalization?

Answer. For CFIC, to induce the intended uniform clustering, micropillars should be arranged at uniform period in every direction. Hence, most micropillars with the micropillars arranged at uniform period are always successfully clustered if there were no unintended particles between them, but micropillars at the very edges of the pattern area that no longer have the arranged pillars always fail to make the intended clustering. By cutting off the edges after the fabrication with a sufficiently large area, only the pattern area with the intended clustering can be obtained.

For chemical coatings for complex wettability were simply performed manually in this work and were always successful enough that there was no need to measure the success rate.

Q 4: Line 146: the authors state that the appropriate design point satisfying all the conditions for effective CFIC should be slightly above the clustering region critical condition where the angle with the water-air interface is minimum (line 144). Anyway, this seems in contrast with figure 2f where the angle is minimum far from the clustering region critical condition.

Answer. The angle of Fig. 2f is the clustering angle of the micropillar bent by capillary force. The higher this value, the lower the angle of contact the clustered pillars would make with the water surface. To deliver the

concept more clearly, the sentences in lines 151 ~ 157 were edited.

Q 5: Line 189: the authors should explain in a better way how they performed the chemical functionalization. Moreover, how long does it last?

Answer. For superhydrophobic coating, wax solution was applied to the surface of the sample and dried, and then excess residual wax was brushed off with compressed air.

For hydrophilic tip coating, the clustered pillar tips were brought into contact with the DA solution and left for 24 hours.

*In the test environment, changes in wettability could not be observed, and it is expected to be maintained semi-permanently without active physical damage. The relevant context is given in the **Methods** section. (Line 296 ~ 299)*

Q 6: Authors state that their strategy is promising for the practical commercialization of the Salvinia-inspired HDR surfaces: is the high aspect ratio of the micropillars achievable also with other microfabrication techniques or is it somehow “limited” by the employment of two-photon lithography?

Answer. In this study, 2PP lithography was used for the accurate and fast fabrication of the master mold required for the experiment. Since micropillars of similar scale can be fabricated without difficulty with conventional micro-fabrication techniques such as photolithography and soft lithography, the employment of 2PP lithography is not necessary. Once the master mold is fabricated, the micropillar array template for CFIC can be easily prepared by conventional soft lithographic replication processes.

Some more aspects need to be clarified: here below are a few recommendations that authors can consider to revise:

Q 7: Line 135: the angle at the clustered tips should be illustrated in a picture in order to clarify this concept.

*Answer. To deliver the concept more clearly, clustered angle φ_0 is illustrated in **Figure 2d**.*

Q 8: Figure 4: the phases of the fabrication process should be illustrated rather than, or not only, the images of the results.

*Answer. Schematic illustration of every phase of the fabrication process was added in **Supplementary Fig. 4**.*

Q 9: Line 250: since the authors state about the gentle curvature of the structures, did they measure it?

*Answer. The average clustering angle of micropillars were measured to be $\sim 54.3^\circ$ with standard deviation of $\sim 1.5^\circ$, which is in a good agreement with the theoretical expectation from **Equation (12)**. Relevant statement was added in lines 216 ~ 218.*

Q 10: Equation 4,5,6,7: the subscript letters of the geometrical parameters should be better explained before the equations.

Answer. To deliver meaning of parameters more clearly, lines 92 ~ 110 were edited.

Q 11: Equation 15: it should be reported later in the text.

*Answer. Relevant lines 244 ~ 245 were edited to refer the **Equation (15)***

Reviewer #2:

This is a beautiful piece of work, well conceived and well worthy of publication.

My few criticisms/comments are as follows.

Q 1: Line 92 et seq.

The description of Equations (1) – (5) is unclear. Although references are given, there is insufficient explanation in the context.

Equation (1) describes self buckling due to gravity. This seems to be basically Euler's formula for the critical load leading to buckling of the beam under its own weight. It would seem to suggest that buckling will not (usually) occur due to gravity, but will, with an applied force and sufficiently large (h/d). However, consideration of polymer surface tension may be equally relevant at this micro-scale. Any bending of the pillar, assumed for simplicity to be a circular cylinder, will induce an effective resistance due to surface tension, γ , of order $\gamma\pi d$. The two effects, elastic and capillary, should be approximately additive and so using the Euler expression, where $I = \pi d^4/64$, the buckling force should be of order $P = \pi^2 EI/h^2 + \pi\gamma d$, where the two components are of similar magnitude for the given system.

Following this, Equation (2) and (3) describe 'ground collapse' and 'lateral collapse'. This is unclear, at least to me. (Although I do note the presence a surface tension term in both.)

Answer. To induce the intended uniform clustering of micropillars, it is essential to maintain a uniform arrangement of micropillars as designed. If the spacing arrangement of the micropillars is disrupted due to buckling or any type of collapse, it is not possible to expect uniform clustering of all pillars through CFIC.

Equations (1), (2), and (3) are shown to find conditions under which the micropillars can remain upright against three different types of structural failure (self-buckling, ground collapse, and lateral collapse) before applying the liquid.

Equation (1) describes self-buckling due to gravity, referred to as 'gravity-induced buckling'.

Equation (2) describes the conditions where the pillar is attached to the ground due to adhesion and cannot restore to its original position, referred to as 'ground collapse'.

Equation (3) describes the conditions where the pillar is attached to adjacent pillar due to adhesion and cannot restore to its original position, referred to as 'lateral collapse'.

Equations (1), (2), and (3) all show conditions for the micropillars to be maintained in an upright state with uniform period before the liquid for CFIC is applied (thus without considering capillary force of the liquid yet).

One of the three collapse types will eventually occur if the aspect ratio of micropillars increases, but which type of collapse will occur first depends on not only the aspect ratio but also various parameters such as the distance to adjacent pillar, surface energy, and modulus of the material. To set the upper design limit in Fig. 2f to confine the desirable design region, a collapse type given micropillars are most susceptible to should be specified.

Obtained from Equations (1) and (3), Equation (4) represents the critical condition in which micropillars become more susceptible to either self-buckling or lateral collapse.

Obtained from Equations (2) and (3), Equation (5) represents the critical condition in which micropillars become more susceptible to either lateral collapse or ground collapse.

Equations (6) and (7) represents the simplified critical conditions by substituting material properties of PDMS into Equation (4) and (5).

From Equations (6) and (7), which collapse type should be considered can be specified as shown in Fig. 2e and Supplementary Fig. 1.

In confining the desirable design region in Fig. 2f, only lateral collapse was considered as the upper limit because the lateral collapse is the most susceptible collapse type for given material and geometry.

To deliver the concept more clearly, relevant lines were edited. (Line 92 ~ 120)

Q 2: Line 174 et seq.

There is a description of the fabrication process mentioning the ‘PDMS precursor’, ‘...negative PDMS stamp, the second positive PDMS replica was demolded with the same typical process. PVA solution was cast on the as-prepared positive PDMS sample and then evaporated...’, etc. For me, at least, it is unclear what polymer the final structure is made of!

Answer. The final structure is mainly made of PDMS, with hydrophobic wax coating over the entire surface and PDA coating at the ends of the clustered pillars. PVA solution was used to induce the clustering of PDMS pillars. Overall surface of the PDMS was coated with hydrophobic wax, and the tips of the clustered pillars were coated with hydrophilic PDA. To deliver the concept more clearly, more detailed explanation was added in lines 188 ~ 193 and in the **Methods** section, lines 280 ~ 285, and a schematic illustration of the fabrication process is added in **Supplementary Fig. 4**.

Q 3: The authors used polyvinyl alcohol (PVA) solution as a liquid to induce the capillary interaction force.’...Why specifically PVA?

Answer. PVA was used because the thin PVA film formed after complete evaporation of the solution could temporarily fix the ends of the clustered pillars together. In this work, a low concentration PVA solution was applied to induce the clustering of micropillars, and at the same time, to temporarily fix the clustered pillars together by the very thin PVA film formed at the ends of the clustered pillars. If pure water is used instead, the clustered pillars simply return to their original position after the evaporation. Other than this, PVA is easy to use and rinse since the PVA is water soluble. The relevant context is in lines 86 ~ 91 and lines 194 ~ 200.

Q 4: Line 209 et seq.

An elliptic approximation is adopted for the meniscus shape in Equation (13). Presumably this is for simplicity, but the solution for such a problem has been known for years. The effective ‘weight’ of the meniscus, which amounts to the pull-off force, is simply $C \cdot \sigma \cos \theta$, where C is the circumference of contact, σ liquid surface tension, and θ the local contact angle. See for example: Dynes and Kaelble (J. Adhesion, 1974), and Schultz et al. (J. Adhesion, 1981). Admittedly, in the present case, it is difficult to determine either C or θ ! However, provided the solid is homogeneous, axially circular and its axis normal to the free liquid surface (Alimov and Kornev (Proc. Roy.Soc. A, 2014)), the FORM of the meniscus is independent of the solid. See, for example, White and Tallmadge (J. Fluid Mech., 1965).

Answer. As the reviewer pointed out, the elliptic approximation in **Equation (13)** was adopted for simplicity, following Gandyra et al. (Beilstein J. Nanotechnol., 2015). For theoretically more accurate calculation, the solution the reviewer kindly mentioned are correct. However, as the reviewer also mentioned, it is difficult to determine the exact local contact angle θ from the microscopic image just before the snap-off. When calculated with rough approximate measurements (thus with high deviation), the range of the pull-off force is approximately 18.1 ~ 20.0 μN (calculated with $C = 295.06 \mu\text{m}$, $\sigma = 72 \text{ mN/m}$, $0.35 < \theta < 0.55$), that is, in a similar order with the value calculated from the elliptic approximation (19.9 μN).

To deliver more comprehensive content, relevant lines were added in lines 233 ~ 235, and under **Supplementary Fig.7**.

A couple of minor points:

Q 5: Supplementary Information, Figure 5, cites contact angle as 168.3°. Are 4 significant figures warranted, especially with no estimate of errors? Also, do the authors know what this contact angle corresponds too, since the drop is ‘sitting’ on an inhomogeneous bed of pillars, or to coin Quéré (Nature Materials, 2002), it is a ‘Fakir droplet’.

Answer. To provide more accurate contact angle of a droplet on the sample surfaces, multiple measurements were further performed. The average contact angle was found to be 174.9° with standard deviation of 3.2°. All significant figures are warranted. Relevant context was edited in lines 219 ~ 221 and in **Supplementary Fig.6**.

Since the droplet is supported partially by the clustered pillars and partially by the air, the apparent contact angle

corresponds to the effective contact angle that can be describe by the Cassie-Baxter state.

Q 6: There seems to be a typo. in Figure 2. There appears to be no d (as in the legend), but there is a d_t .

Answer. *The typo was corrected, by replacing the d in the caption of **Fig. 2** to d_t .*

REVIEWERS' COMMENTS

Reviewer #1 (Remarks to the Author):

In the present manuscript, the authors propose the fabrication of a biomimetic surface inspired by *Salvinia Molesta* leaves. In particular, they have developed a relatively simple and effective procedure for the fabrication and functionalization of 3D crown-like shapes. This represents a noteworthy result in the field of biomimetic wettability since large-scale production is one of the main limiting factors for technological applications.

The authors properly addressed all my comments. All the aspects have been delivered more clearly, especially regarding the fabrication procedure. All relevant details have been added to the text.

Reviewer #2 (Remarks to the Author):

The paper seems OK to publish to me, apart from 2 points necessary to finalise the work:

(1) Line 246: 'From Equation (15).....136.95 μ m.' I do not believe 5 significant figures are warranted here.

(2) The references that I suggested have been added, granted, but NOT referred to correctly. They should be in the reference list, surely, and cited fully and correctly, which the authors have not made the effort to do.

Although it is not my job to do this for them, rapid consultation of the Web of Science easily yields the full details:

--SURFACE-ENERGY ANALYSIS OF CARBON-FIBERS AND FILMS

By:

DYNES, PJ (DYNES, PJ) ; KAELBLE, DH (KAELBLE, DH)
JOURNAL OF ADHESION, Volume 6, Issue 3, Page 195-206.
DOI 10.1080/00218467408075026
Published 1974

--FIBER SURFACE-ENERGY CHARACTERIZATION

By:

SCHULTZ, J (SCHULTZ, J) ; CAZENEUVE, C (CAZENEUVE, C) ; SHANAHAN, MER (SHANAHAN, MER) ;
DONNET, JB (DONNET, JB)
JOURNAL OF ADHESION, Volume 12, Issue 3, Page 221-231.
DOI 10.1080/00218468108071202
Published 1981

--Meniscus on a shaped fibre: singularities and hodograph formulation

By:

Alimov, MM (Alimov, Mars M.) [1] ; Kornev, KG (Kornev, Konstantin G.) [2]
View Web of Science ResearcherID and ORCID
(provided by Clarivate)
PROCEEDINGS OF THE ROYAL SOCIETY A-MATHEMATICAL PHYSICAL AND ENGINEERING SCIENCES
Volume 470, Issue 2168, Article Number 20140113.
DOI 10.1098/rspa.2014.0113
Published AUG 8 2014

Point-to-point responses to the reviewers' comments

Reviewer #2:

The paper seems OK to publish to me, apart from 2 points necessary to finalise the work:

Q 1: Line 246: 'From Equation (15) ... 136.95 μm .' I do not believe 5 significant figures are warranted here.

Answer. As **Reviewer #2** kindly pointed out, the calculated value should be denoted as 137 μm with 3 significant figures warranted, since h (30.0 μm) have the lowest significant figures (3) in the Equation. The manuscript was edited accordingly.

Q 2: The references that I suggested have been added, granted, but NOT referred to correctly. They should be in the reference list, surely, and cited fully and correctly, which the authors have not made the effort to do.

Answer. The additional references are now cited fully and correctly in the reference list in the main manuscript (#46-48) and the supplementary information (#1-3).